# Cerebral blood flow and histological analysis for the accurate differentiation of infiltrating tumor and vasogenic edema in glioblastoma

Hideki Kuroda[1], Yoshiko Okita[1]*, Atsuko Arisawa[2], Reina Utsugi[1], Koki Murakami[1], Ryuichi Hirayama[1], Noriyuki Kijima[1], Hideyuki Arita[1,3], Manabu Kinoshita[1,4], Yasunori Fujimoto[1,5], Hajime Nakamura[1], Naoki Kagawa[1], Noriyuki Tomiyama[2], Haruhiko Kishima[1]

1 Department of Neurosurgery, Osaka University Graduate School of Medicine, Osaka, Japan,
2 Department of Diagnostic Radiology, Osaka University Graduate School of Medicine, Osaka, Japan,
3 Department of Neurosurgery, Osaka International Cancer Institute, Osaka, Japan, 4 Department of Neurosurgery, Asahikawa Medical University, Asahikawa, Japan, 5 Department of Neurosurgery, Osaka Rosai Hospital, Sakai, Osaka, Japan

* yokita4246@gmail.com

**Data Availability Statement:** Data Availability Statement: The raw data in this study cannot be shared publicly as it includes identifiable

## Abstract

### Background

Glioblastoma is characterized by neovascularization and diffuse infiltration into the adjacent tissue. T2*-based dynamic susceptibility contrast (DSC) MR perfusion images provide useful measurements of the biomarkers associated with tumor perfusion. This study aimed to distinguish infiltrating tumors from vasogenic edema in glioblastomas using DSC-MR perfusion images.

### Methods

Data were retrospectively collected from 48 patients with primary *IDH*-wild-type glioblastoma and 24 patients with meningiomas (Edemas-M). First, we attempted histological verification of cell density, Ki-67 index, and microvessel areas to distinguish between non-contrast-enhancing tumors (NETs) and edema (Edemas) which were obtained from stereotactically fused T2-weighted and perfusion images. This was performed for evaluating enhancing tumors (ETs), NETs, and Edemas. Second, we also performed radiological verification to distinguish NETs from Edemas. Two neurosurgeons manually assigned the regions of interests (ROIs) to ETs, NETs, and Edemas. The DSC-MR perfusion imaging-derived parameters calculated for each ROI included the cerebral blood volume (CBV), cerebral blood flow (CBF), and mean transit time (MTT).

### Results

Cell density and microvessel area were significantly higher in NETs than those in Edemas (p<0.01 and p<0.05, respectively). Regarding radiological analysis, the mean CBF ratio for Edemas was significantly lower than that for NETs (p<0.01). The mean MTT ratio for Edemas was significantly higher than that for NETs. The receiver operating characteristic

information such as the name of the treatment hospital (Osaka University Hospital), diagnostic details, age, gender, and clinical outcomes. The raw data are available upon reasonable request from Osaka University Graduate School of Medicine, Ethics Committee via email (rinri@hp-crc.med.osaka-u.ac.jp) or telephone (+81-6-6210-8296).

**Funding:** This work was supported by JST Grant Number JPMJPF2009.The funders had no role in study design, data collection and analysis, decision to publish, or preparation of the manuscript.

**Competing interests:** The authors have declared that no competing interests exist.

(ROC) analysis showed that CBF (area under the curve [AUC] = 0.890) could effectively distinguish between NETs and Edemas. The ROC analysis also showed that MTT (AUC = 0.946) could effectively distinguish between NETs and Edemas.

## Conclusions

DSC-MR perfusion images may prove useful in differentiating NETs from Edemas in non-contrast T2 hyperintensity regions of glioblastoma.

## Introduction

Glioblastoma is the most common type of malignant brain tumor [1,2]. The prognosis of glioblastoma is extremely poor, with a median survival of 14.6 months, even with standard treatments, such as chemoradiotherapy [3]. The extent of surgical resection is a known prognostic factor for survival in patients with newly diagnosed glioblastoma [4]. Adding a resection-infiltrating tumor beyond the contrast-enhancing margin-preserving function is associated with the possibility of longer survival [5–8]. Owing to the highly infiltrative nature of glioblastoma, systematic reviews have demonstrated a positive correlation between supratotal resection and overall survival in glioblastoma [5–8]. However, the current preliminary results aiming for resection with maximum safety, although promising, provide no clear radiological or anatomical definition regarding the extent of tumor resection that can be performed without the occurrence of new postoperative neurological deficits.

T2-weighted images of regions beyond the contrast-enhancing region in glioblastoma may represent different histopathological conditions, such as vasogenic edema or peritumoral infiltration [9]. In terms of incomplete surgical resection for infiltrating tumors, the peritumoral area predominantly contains infiltrating tumor cells and directly affects patient outcomes [10]. Microvascular proliferation is a crucial histological feature of glioblastoma. Glioblastoma is characterized by active neovascularization surrounded by the tumor border zone with diffuse infiltration into the adjacent brain tissue. T2*-based dynamic susceptibility contrast (DSC) MRI is a clinical imaging technique that non-invasively assesses the microvascular status of glioblastoma [11,12]. Although, several studies have been conducted to compare non-enhancing peritumoral areas of glioblastomas and MR perfusion metrics, the non-enhancing peritumoral was analyzed without distinguishing between infiltrating tumors and edema [13–16]. On the other hand, a few studies have been conducted to discriminate between non-enhancing tumor and perilesional edema using machine learning algorithms with various imaging parameters comprising DSC-MRI. To date, there is still no current consensus on a clinically usable method for reliably differentiating non-enhancing tumor from perilesional edema on perfusion imaging [17–19].

We aimed to assess the characteristics of the non-enhancing peripheral area in glioblastoma, radiologically and pathologically, and differentiate between non-contrast-enhancing tumors (NETs) and vasogenic edema (Edemas) to increase the diagnostic accuracy based on the histopathological differences in these surrounding areas on T2-weighted images using DSC-MR perfusion images.

## Materials and methods

### Study design and patient selection

This retrospective study was approved by the Clinical Research Review Committee of Osaka University (Approval No.: 22302). All participants provided informed written consent. The present research, and all methods contained within, was conducted in accordance with the Declaration of Helsinki. We accessed data from medical records for research purposes between October 18, 2022, and April 28, 2023. During this process, we had access to information that could identify individual participants.

Forty-eight patients with histologically confirmed primary *IDH*-wild-type glioblastoma (according to the 2021 World Health Organization International Histological Classification of Tumors) who underwent tumor resection at our institution between January 2017 and January 2023 were included in the study (male: female, 31: 17; median age, 65.6 years) (Fig 1). DSC-MR perfusion images and conventional MR pulse sequences were acquired for all patients pre-surgically. Peritumoral brain edema in patients with meningioma was defined as vasogenic edema in previous reports [20–22]. We referred to these reports and measured meningioma as a reference subset for comparison of vasogenic edema. Data from 24 patients with WHO Grade1 meningioma who presented evaluable edema in MRI and underwent tumor resection after preoperative embolization between January 2019 and January 2023 were also collected as control data (male: female, 10: 14; median age, 63.0 years) (Edemas-M). They also underwent DSC-MR perfusion imaging and conventional MR pulse sequence pre-embolization.

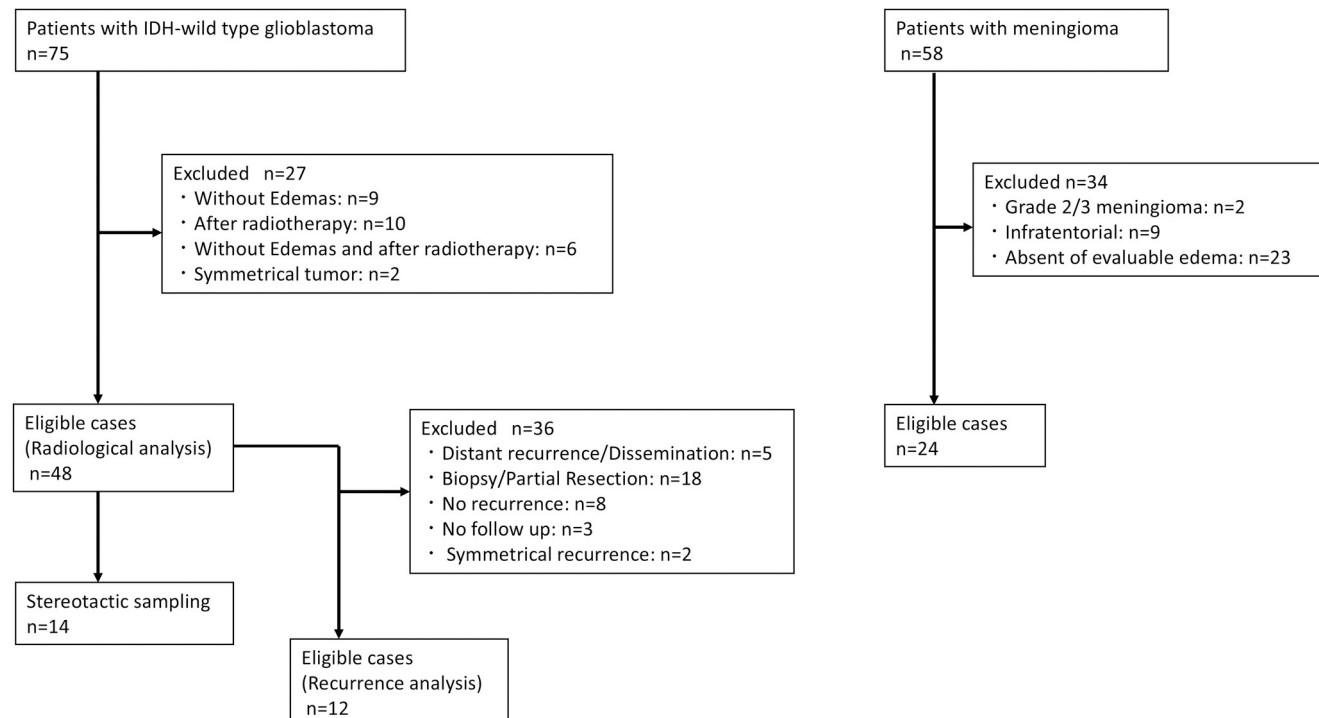

**Fig 1. The process of patient selection for inclusion in the study.** We retrospectively collected the medical records of 75 patients diagnosed with *IDH*-wildtype glioblastoma between January 2017 and January 2023 at our institution. Of those 75 patients, 27 were excluded from the study due to tumors with unsatisfactory images and/or prior radiotherapy before MRI examination, leaving 48 eligible for radiological analysis. In 14 of these cases, stereotactic sampling was performed. During the review period, local recurrent analysis was also conducted. Of the 48 patients, 36 were excluded from the recurrent analysis due to recurrent tumors with unsatisfactory images, biopsy or partial resection, lack of recurrence, or no follow-up. Additionally, we retrospectively collected the medical records of 58 patients diagnosed with meningioma between January 2017 and January 2023 at our institution. Of those 58 patients, 38 were excluded from the study due to tumors with unsatisfactory images, Grade 2/3 meningioma, or infratentorial tumors.

## Magnetic resonance imaging

All images, including axial T1-, T2-, T2*-weighted images, Fluid Attenuated Inversion Recovery, contrast-enhanced T1-weighted sequence (T1Gd), and diffusion tensor imaging (DTI) were obtained using a 3-T MR unit (DISCOVERY MR 750, GE Healthcare, Milwaukee, WI, USA) with a 32-channel head coil. DSC-MRI was performed using gradient-echo EPI (GRE-EPI) during contrast agent administration. The imaging parameters were: 2000-millisecond TR/20.9-millisecond TE, 60° flip angle, 3906.2-Hz pixel bandwidth, 220 × 220mm FOV, 1.719 × 1.719 × 5-mm voxel size, and a 1-mm interslice gap. Considering each section, 40 images were obtained at intervals equal to the TR. After approximately 8 time points, 0.1 mmol/kg of meglumine gadoterate (Guerbet Japan, Tokyo, Japan) was injected at a rate of 3 mL/s, immediately followed by a 30-mL saline flush. DTI was acquired using a single-shot echo planar imaging technique with TE = 80 and TR = 10,000. Diffusion gradient encoding in 25 directions with b = 2,000 s/mm2 and an additional measurement without the diffusion gradient (b = 0 s/mm2) was performed [23].

## Image postprocessing

The images were postprocessed using a dedicated software package Synapse Vincent (Fuji Medical Systems, Tokyo, Japan). MRI perfusion images were analyzed qualitatively using rainbow color scale maps of cerebral blood flow (CBF), cerebral blood volume (CBV), and mean transit time (MTT). Next, postprocessing of the acquired images into CBF, CBV, and MTT maps was performed quantitatively. DTI was also acquired qualitatively apparent diffusion coefficient (ADC) values using an ADC map. This software used the deconvolution method as previously reported [24–27].

## Imaging analysis

Imaging analysis was performed using Synapse Vincent in the perfusion mode. Two neurosurgeons (Hi.K. and Y.O. with 8 and 21 years of experience in neuroradiology, respectively) discussed and classified the non-enhancing T2-weighted hyperintense area into NETs, namely infiltrating tumors, and Edemas based on a previous report [28] using morphological MRI features. NETs are characterized by gray matter involvement, eccentricity, relatively mild T2 hyperintensity, and focal parenchymal expansion, whereas Edemas are characterized by spared gray matter, relative concentricity around enhancing lesions, marked T2 hyperintensity, and a more diffuse mass effect in cases of marked Edemas [28].

A region of interest (ROI) of 2 mm diameter, comparable to the diameter of the biopsy tissue, was set manually in each of the Edema, NET, and enhancing tumor (ET) regions, and the values of cerebral blood volume (CBV), cerebral blood flow (CBF), mean transit time (MTT), and ADC were calculated. The ROI of identical size was also set manually in contralateral normal area for each ROI for Edema, NET, and ET. The diseased-to-normal ratios were calculated by dividing the values of CBV, CBF, MTT, and ADC for the NET, Edema, and ET by those values of contralateral normal area (Fig 2). At the maximum one, ROI of each type was placed in each of the 48 patients with glioblastoma.

## Surgery for stereotactic multiple sampling evaluation

The location for the tumor biopsy was determined preoperatively on the contrast-enhanced T1-and T2-weighted images. The contrast-enhanced T1- and T2-weighted images were transferred to Brainlab (Brainlab, München, Freistaat Bayern, Germany) or Stealthstation (Medtronic, Dublin, Ireland), and the biopsy target for histopathological examination was planned.

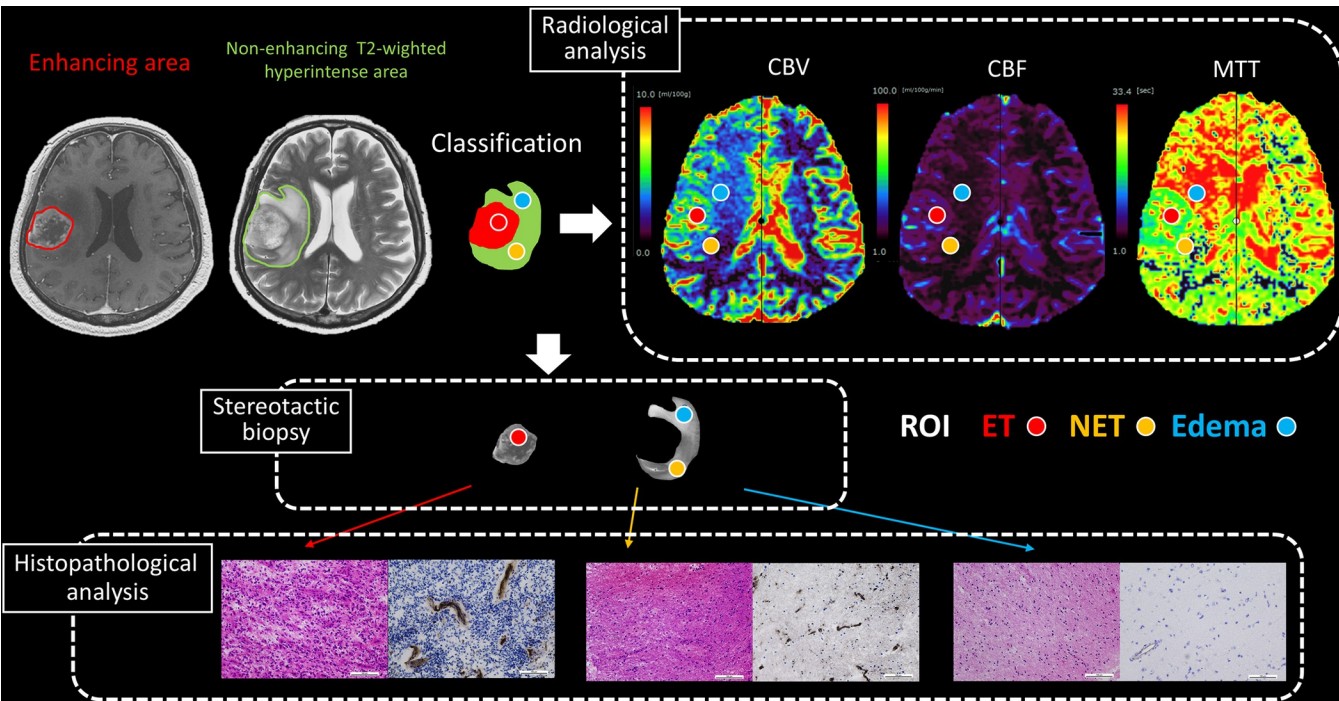

**Fig 2. Illustration depicting image processing.** Representative images featuring contrast-enhanced T1-weighted and non-contrast-enhanced T2-weighted images of a non-contrast-enhancing tumor (NET), edema (Edema), and an enhancing tumor (ET). The region of interest (ROI) is selected to show the cerebral blood volume (CBV), cerebral blood flow (CBF), and mean transit time (MTT) on fused non-contrast-enhanced T2-weighted images. Stereotactic biopsy of each ROI is guided by a neuronavigation system. The cell density, Ki-67 index, and microvessel area are evaluated at the exact ROI in each biopsy specimen.

We used a stereotactic multiple sampling evaluation for gliomas as previously described [29]. The location of tumor biopsy was determined in a similar manner to open biopsy. Standard craniotomy was performed under general anesthesia in all patients. Multiple sampling biopsy was performed on ETs, NETs, and Edemas, targeted for resection immediately after craniotomy to minimize the error caused by brain shifting. Although multiple tissue samplings were performed in some cases, real-time navigation was performed to confirm the position of each biopsy site (Table 1).

## Histopathological analysis

Hematoxylin and eosin staining and immunohistochemistry (IHC) were performed in all cases. The tumor samples were fixed in buffered formalin, embedded in paraffin or fixed in sucrose, and encapsulated in an optimal cutting temperature compound. The blocks were sectioned into 6-μm tissue sections. Deparaffinized, hydrated with graded alcohol, and heat-activated antigen activation were performed as needed. After blocking the endogenous peroxidase activity, the tissue was incubated with the primary antibodies Ki-67 (mouse monoclonal antibody; clone MIB-1; DAKO; 1:100) and CD31 (mouse monoclonal antibody; clone JC70A; Dako; ready to use). Positive immunostaining was demonstrated with the diaminobenzidine reaction, and slides were subsequently counterstained with hematoxylin, dehydrated, cleared and mounted. The prepared sections were examined in a 200× field of view.

Cell counting was performed under a light microscope (Olympus, Tokyo, Japan) at 200× magnification. The area for the tumor cell count was per field quantified using ImageJ 1.53k (National Institutes of Health, Bethesda, ML USA) in approximately three areas at 200x magnification, and data was recorded as the mean of three different locations within the specimen.

**Table 1. Patient characteristics with stereotactic tissue sampling.**

| # | Sex | Age | MGMT promoter metylation status | TERT promoter mutation | Tumor location | Enhanced tumor volume (cm3) | Number of stereotactic tissues | | |
|---|---|---|---|---|---|---|---|---|---|
| | | | | | | | ET | NET | Edema |
| 1 | M | 65 | unmethylated | mutant | Right parietal | 20.7 | 0 | 1 | 1 |
| 2 | M | 86 | metylated | mutant | Left temporal | 0.54 | 0 | 1 | 1 |
| 3 | M | 81 | metylated | wild type | Right frontal | 92.1 | 1 | 1 | 1 |
| 4 | M | 74 | metylated | mutant | Right temporal | 10.4 | 1 | 1 | 0 |
| 5 | F | 79 | metylated | wild type | Left frontal | 63.1 | 0 | 1 | 1 |
| 6 | F | 71 | unmethylated | wild type | Left occipital | 8.2 | 1 | 1 | 1 |
| 7 | M | 78 | unmethylated | mutant | Right parietal | 80.7 | 1 | 0 | 1 |
| 8 | M | 52 | unmethylated | mutant | Left frontal | 45.3 | 1 | 1 | 1 |
| 9 | M | 55 | unmethylated | mutant | Right parietal | 12 | 1 | 1 | 1 |
| 10 | F | 68 | unmethylated | mutant | Right temporal | 1.21 | 1 | 1 | 0 |
| 11 | F | 88 | unmethylated | wild type | Right temporal | 63 | 1 | 1 | 1 |
| 12 | F | 72 | unmethylated | mutant | Right basal ganglia | 25 | 1 | 1 | 0 |
| 13 | F | 84 | unmethylated | wild type | Left temporal | 13.3 | 1 | 1 | 0 |
| 14 | M | 50 | metylated | mutant | Left temporal | 74.8 | 1 | 0 | 0 |

ET = enhancing tumor, NET = non-contrast-enhancing tumors, Edema = vasogenic edema.

Ki-67 labeled cells were also counted and the percentage of Ki-67 labeled cells was calculated within the observed field. In addition, the tumor microvessel area was determined by calculating the area of the lumens and walls of tumor microvessels occupying the tissue per field quantified using ImageJ 1.53k in approximately three areas at 200x magnification. The average values obtained from approximately three fields were recorded as the value of the microvessel area.

## Statistics analysis

Statistical analysis was performed using JMP software (version 16.0; SAS Institute, Cary, NC, USA). The Mann–Whitney U-test was used for comparisons between two groups, whereas the Steel–Dwass test was used for comparisons between three or more groups. Nonparametric tests were performed due to the small sample size. Spearman correlation was used for evaluating correlations between continuous variables. Receiver operating characteristic (ROC) curve analysis was performed to compare the performance of each imaging parameter based on each ROI for distinguishing NETs from Edemas. The p-values were considered statistically significant at $p < 0.05$.

## Results

### Correlation of MR perfusion imaging parameters and ADC values with Ki-67 index, cell density, and microvessel area values based on a histological comparison using stereotactic imaging

Based on a previous report that used morphological features, ROIs were placed regarding 12, 9, and 11 patients with NETs, Edemas, and ETs, respectively, in glioblastomas [28]. Stereotactic tissue specimens in the areas that corresponded to the pre-defined ROIs of 12, 9, and 11 patients with NETs, Edemas, and ETs, respectively, were also obtained via post-surgical histological examination (Table 2). The Ki-67 index, cell density, and microvessel area were

**Table 2. The average of all parameters of ROIs in ETs, NEsT, Edemas, and Edemas-M.**

| | Radiological analysis | | | | | | Histopathological analysis | | | | | |
|---|---|---|---|---|---|---|---|---|---|---|---|---|
| | CBV ratio | p | CBF ratio | p | MTT ratio | p | Ki-67(%) | p | Cell density | p | Maicrovessel area(μm2) | p |
| ETs | 1.31 | | 2.15 | | 0.7 | | 15.28 | | 1010 | | 14362.6 | |
| | | 0.97 | | 0.52 | | <0.01 | | 0.02 | | 0.13 | | 0.01 |
| NETs | 1.32 | | 1.64 | | 0.96 | | 5.92 | | 625.8 | | 5875.4 | |
| | | 0.1 | | <0.01 | | <0.01 | | 0.02 | | <0.01 | | 0.04 |
| Edemas | 1.01 | | 0.56 | | 1.83 | | 1.87 | | 291.6 | | 2122.4 | |
| | | 0.68 | | 0.65 | | 0.77 | | | | | | |
| Edemas-M | 1.18 | | 0.7 | | 1.75 | | | | | | | |

ET; enhancing tumor, NET; non-contrast-enhancing tumor,Edemas; vasogenic edema, Edemas-M; meningioma, CBV; cerebral blood volume, CBF; cerebral blood flow, MTT; mean transit time.

significantly higher in NETs than those in Edemas (p<0.05, p<0.01, and p<0.05, respectively) (Fig 3A–3C). The Ki-67 index and microvessel area were significantly higher in ETs than those in NETs (p<0.05, and p<0.05, respectively). However, no significant difference was observed in cell density between ETs and NETs.

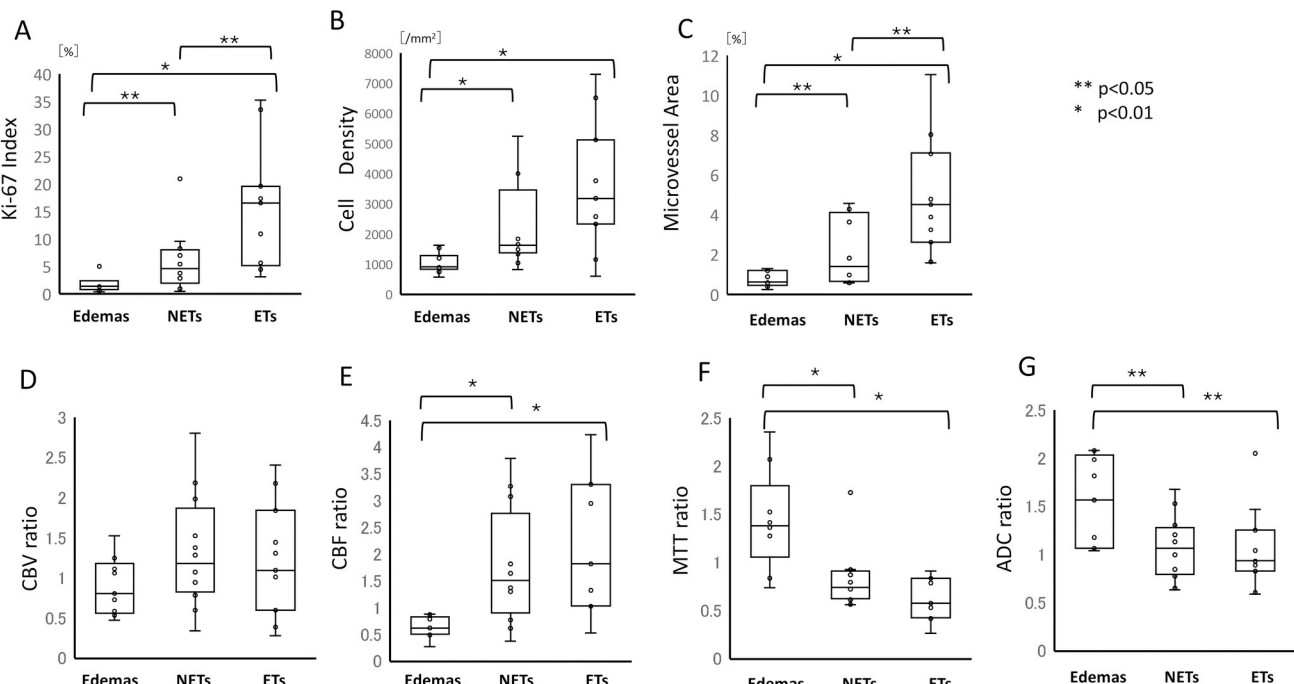

**Fig 3.** A, B, and C. Boxplots of the Ki-67 index, cell density, and microvessel area in non-contrast-enhancing tumors (NETs), edemas (Edemas), and enhancing tumors (ETs). The Ki-67 index, cell density, and microvessel area were significantly higher in NETs compared to Edemas (p<0.05, p<0.01, and p<0.05, respectively). The Ki-67 index and microvessel area were significantly higher in ETs than in NETs (p<0.05 and p<0.05, respectively). However, no significant difference in cell density between ETs and NETs was observed. D, E, F, and G. Boxplots of the cerebral blood volume (CBV), cerebral blood flow (CBF), mean transit time (MTT), and apparent diffusion coefficient (ADC) ratios in non-contrast-enhancing tumors (NETs), edemas (Edemas), and enhancing tumors (ETs) based on a histological comparison using stereotactic imaging. The mean CBF ratio for Edemas was significantly lower than that for NETs (p<0.01). The mean MTT ratio for Edemas was significantly higher than that for NETs (p<0.01) The ADC ratios were significantly higher in NETs than in Edemas (p<0.05). However, no significant difference was observed in CBV ratios between NETs and Edemas.

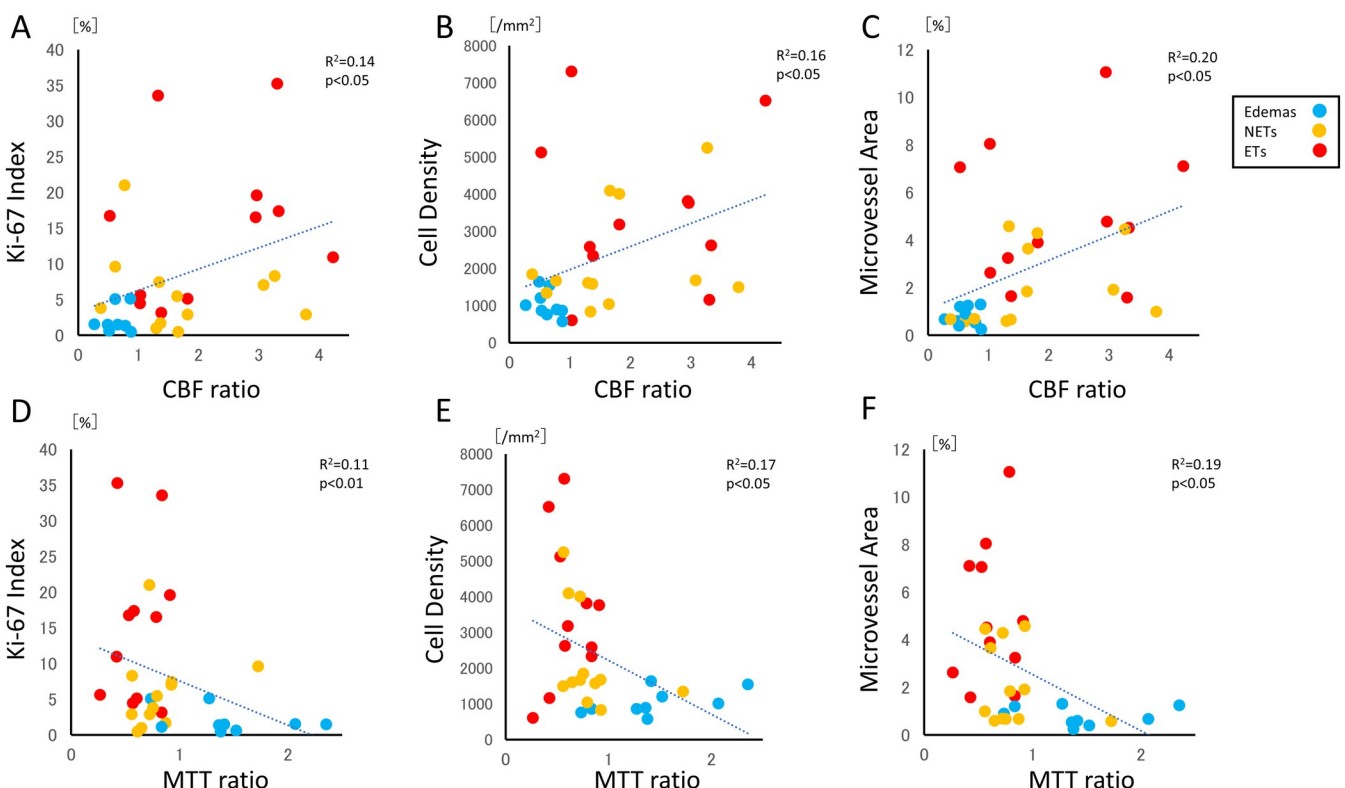

**Fig 4. Correlation of CBF ratio with cell density, Ki-67 and microvessel area based on a histological comparison using a stereotactic image.** The CBF ratio shows correlation with cell density (A: R = 0.400, p = 0.023), Ki-67 index (B: R = 0.374, p = 0.034), and microvessel area (C: R = 0.443, p = 0.011). Correlation of MTT ratio with cell density, Ki-67 and microvessel area based on a histological comparison using a stereotactic image. The MTT ratio shows correlation with cell density (D: R = 0.409, p = 0.02), Ki-67 index (E: R = 0.322, p = 0.0003), and microvessel area (F: R = 0.430, p = 0.014).

No significant difference was observed in the mean CBV ratios between Edemas, NETs, and ETs (Fig 3D). The mean CBF ratio for Edemas was significantly lower than that for NETs (p<0.01). In contrast, the mean MTT ratio for Edemas was significantly higher than that for NETs (p<0.01). However, no significant difference was observed in the CBF and MTT ratios between ETs and NETs (Fig 3E and 3F). To further clarify the distinction between NETs and Edemas, we compared the validation using ADC values. The ADC ratios were significantly higher in NETs than in Edemas (p<0.05) (Fig 3G). However, no significant difference was observed in ADC ratios between ETs and NETs.

Stereotactic local comparison of CBF ratio with cell density, Ki-67 index, and microvessel area were performed. The CBF ratio showed a correlation with cell density (R = 0.400, p = 0.023), Ki-67 index (R = 0.374, p = 0.034), and microvessel area (R = 0.443, p = 0.011), respectively (Fig 4A–4C). The MTT ratio also showed a correlation with cell density (R = 0.409, p = 0.02), Ki-67 index (R = 0.322, p = 0.0003), and microvessel area, respectively (R = 0.430, p = 0.014) (Fig 4D–4F). However, CBV ratio has shown no correlation with cell density, Ki-67 index, or microvessel area. Similarly, stereotactic local comparisons of the ADC ratio with cell density, Ki-67 index, and microvessel area revealed no correlations among these factors (Fig 5A–5C).

We visually represented the interrelationships between cell density, Ki-67 index, and micro-vascular area in stereotactically evaluated ETs, NETs, and Edemas using a 3D scatter plot (Fig 6). The plots for Edemas were distributed over a smaller area for cell density, Ki-67 index,

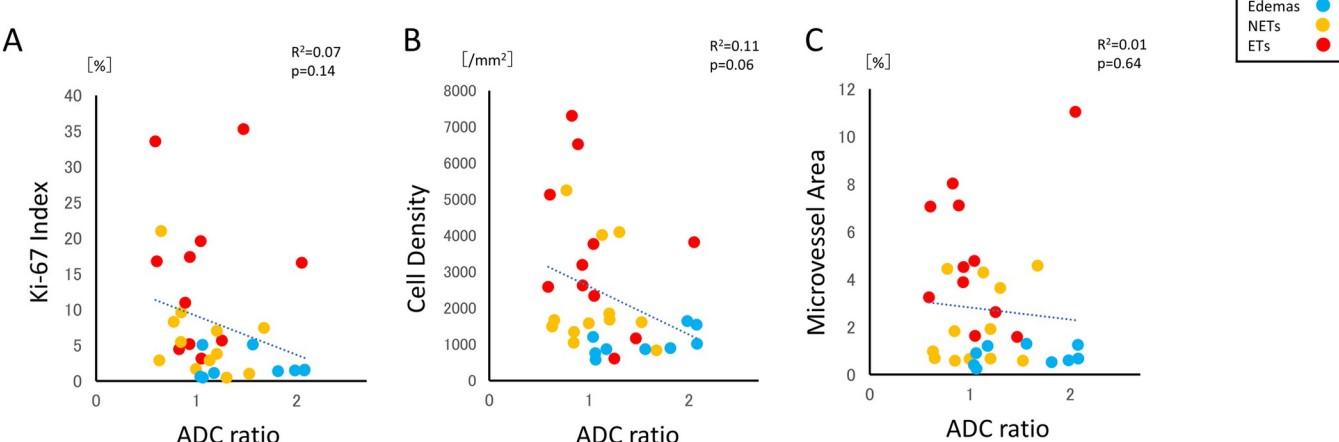

**Fig 5. Correlation of the ADC ratio with cell density, Ki-67, and microvessel area based on a histological comparison using stereotactic imaging.** The ADC ratio shows no correlation with cell density, the Ki-67 index, or microvessel area.

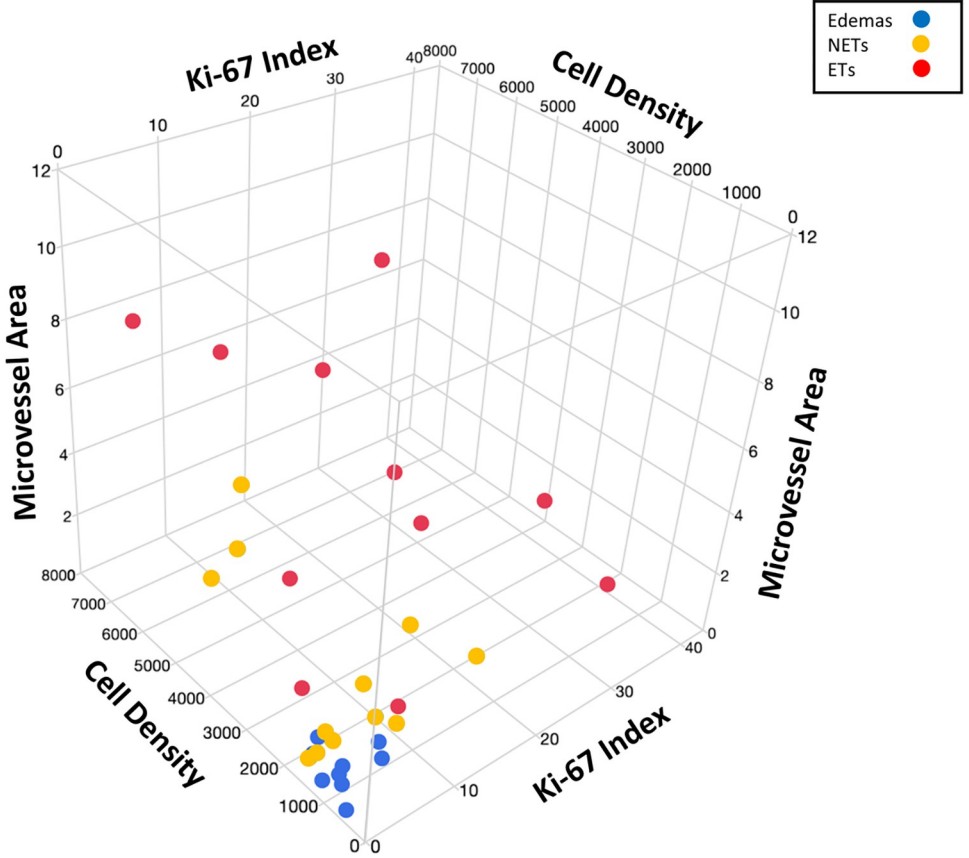

**Fig 6. 3D scatter plot representing the interrelationships between cell density, the Ki-67 index, and microvascular area in stereotactically evaluated ET, NETs, and Edemas.** The plots for Edemas were distributed in a smaller area for cell density, the Ki-67 index, and microvascular area compared to those for NETs and ETs. Although the plots for NETs and ETs were mixed, the plots for ETs were found in areas with higher values for cell density, the Ki-67 index, and microvascular area compared to the plots for NETs.

and microvascular area compared to those for NETs and ETs. Additionally, while the plots for NETs and ETs overlapped, the plots for ETs were located in areas with higher values for cell density, Ki-67 index, and microvascular area than those for NETs.

## Diagnostic value of MR perfusion imaging parameters in differentiating between NETs and Edemas

Based on a previous report that used morphological features, 31, 35, and 40 ROIs were placed on NETs, Edemas, and ETs, respectively, in glioblastomas [28]. Similarly, 24 ROIs were placed on the edema of meningiomas as controls. The mean CBF ratio for Edemas (0.56; range, 0.11–1.08) was significantly lower than that for NETs (1.64; range, 0.39–4.05) (p<0.01). The mean MTT ratio for Edemas (1.83; range, 1.22–3.30) was significantly higher than that for NETs (0.96; range, 0.34–1.60) (p<0.01). In contrast, the CBF and MTT ratios for Edemas and the controls showed similar tendencies (Fig 7, Table 2).

The receiver operating characteristic (ROC) analysis showed that the CBF ratio (area under the curve [AUC] = 0.890) could effectively distinguish between NETs and Edemas with a sensitivity and specificity of 74.2% and 97.1%, respectively (cut-off value = 0.943, p<0.01). The ROC analysis also showed that MTT (AUC = 0.946) was effective in distinguishing between NETs and Edemas with a sensitivity and specificity of 80.6% and 97.1%, respectively (cut-off value = 1.229, p<0.01) (Fig 8).

We presented the probability map applied to NETs and Edemas lesions in two patients with glioblastoma and meningioma (Figs 9 and 10). The probability map predicting NETs indicated

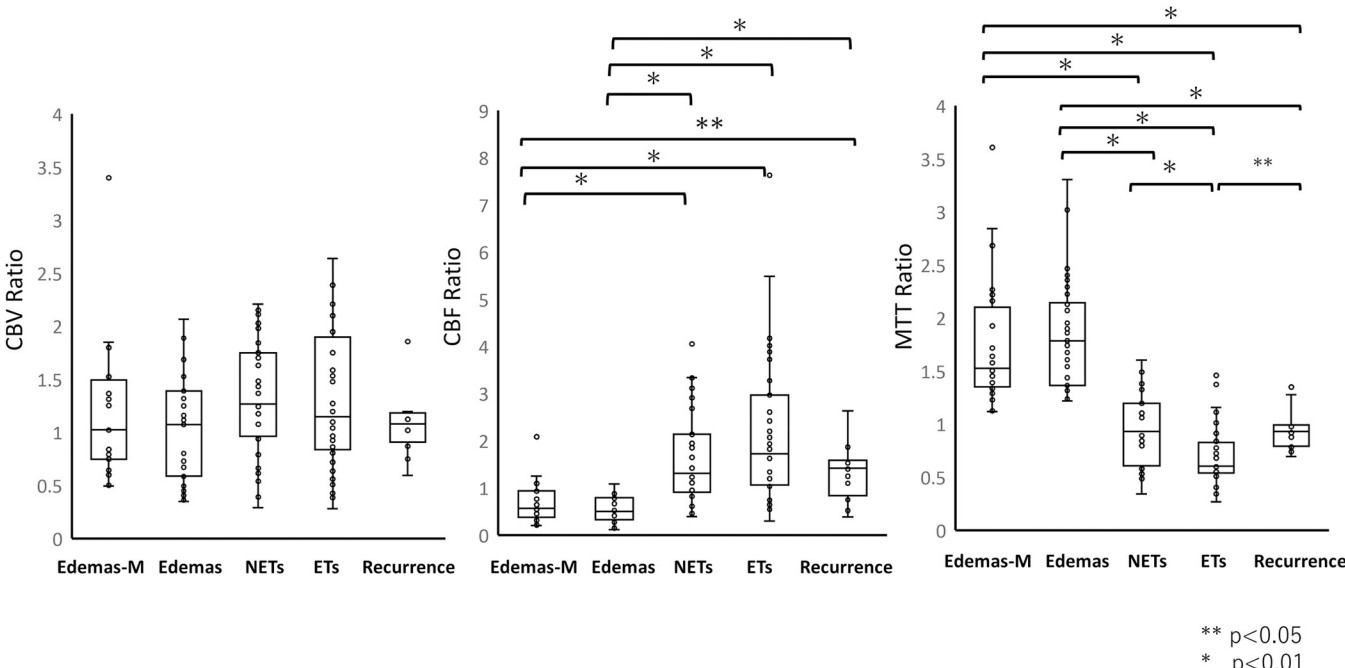

**Fig 7. Boxplots of the cerebral blood volume (CBV), cerebral blood flow (CBF), and mean transit time (MTT) ratios in non-contrast-enhancing tumors (NETs), edemas (Edemas), enhancing tumors (ETs) in 48 glioblastoma patients, and Edemas in the control group (Edemas-M).** The mean CBF ratio for Edemas (0.56; range, 0.11–1.08) was significantly lower than that for NETs (1.64; range, 0.39–4.05) (p<0.01). The mean MTT ratio for Edemas (1.83; range, 1.22–3.30) was significantly higher than that for NETs (0.96; range, 0.34–1.60) (p<0.01). In contrast, the CBV and MTT ratios for Edemas and the controls exhibited similar tendencies. Boxplots of the cerebral blood flow (CBF) and mean transit time (MTT) ratios in local recurrent tumors in 12 relapsed glioblastoma patients. The mean CBF ratio for the recurrent tumor was 1.136, exceeding the cut-off value of 0.943 to predict NETs. The mean MTT ratio for the recurrent tumor was 0.944, falling below the cut-off value of 1.229 to predict NETs.

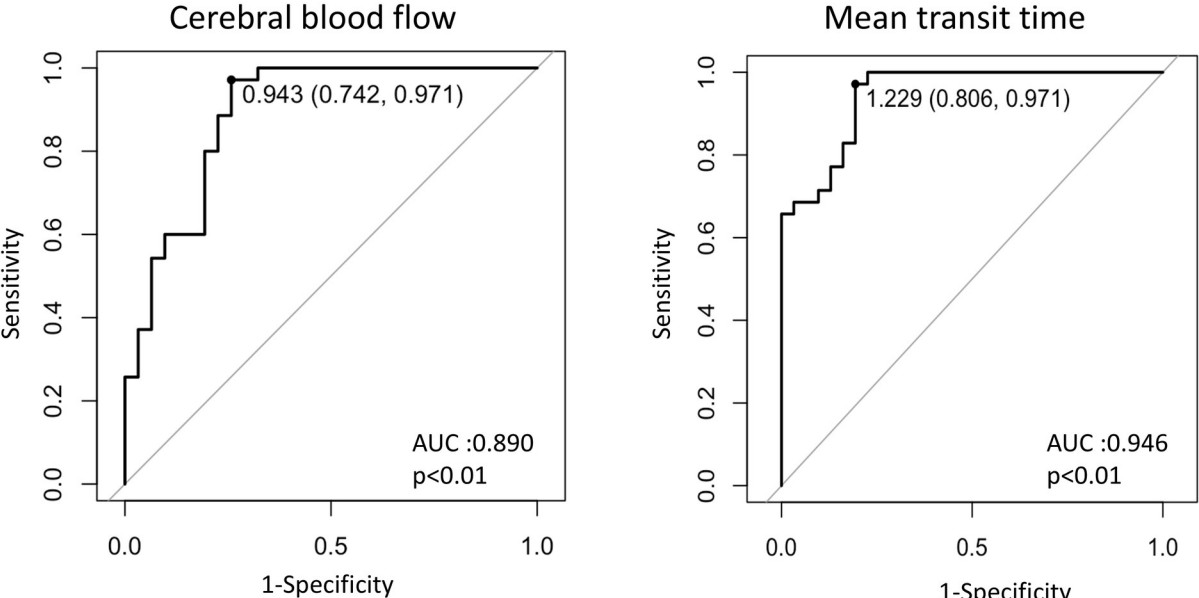

**Fig 8. The receiver operating characteristic (ROC) curve shows reliable predictions that distinguish non-contrast-enhancing tumors (NETs) and edemas (Edemas) in glioblastoma in terms of cerebral blood flow (CBF) and mean transit time (MTT).** The CBF ratio (area under the curve [AUC] = 0.890) effectively distinguishes between NETs and Edemas, with a sensitivity of 74.2% and a specificity of 97.1% (cut-off value = 0.943, p<0.01). The MTT ratio (AUC = 0.946) also effectively distinguishes between NETs and Edemas, with a sensitivity of 80.6% and a specificity of 97.1% (cut-off value = 1.229, p<0.01).

a CBF ratio over 0.943, represented in orange. The area colored blue corresponded to Edema CBF ratios below 0.943. Additionally, the probability map predicting NETs showed an MTT ratio below 1.229, also colored orange, while the area colored blue applied to Edema MTT ratios above 1.229. Notably, the area of the probability CBF map for Edemas containing NETs did not necessarily align with the area in the probability MTT map for Edemas in both glioblastoma and meningioma cases.

### Prognostic values of MR perfusion imaging parameters

We evaluated the 12 local recurrent cases for preoperative CBF ratios and MTT ratios at recurrent lesions. All lesions were applicable in NETs and turned into enhancing tumors at recurrence. The mean CBF ratio for the recurrent tumors was 1.136, above the cut-off value of 0.943 to predict NETs. The mean MTT ratio for the recurrent tumors was 0.944, below the cut-off value of 1.229 to predict NETs (Figs 7 and 8).

### Discussion

Our study indicates that CBF and MTT can be used to distinguish NETs from Edemas in glioblastoma using stereotactic histological and radiological analyses. We performed histological analysis of the microvessel areas, which were identified using stereotactically fused T2-weighted and DSC-MR perfusion images, for ETs, NETs, and Edemas. Noguchi et al. proposed that ASL-driven CBF may predict the histopathological vascular densities of brain tumors [30]. Furthermore, Ningning et al. suggested that ASL-driven CBF showed a statistically significant positive correlation with microvascular density of tumor core lesions stereotactically biopsied from patients with gliomas [31]. In contrast, Rotkopf et al. investigated the correlation between DSC-MR perfusion imaging and stereotactic histological vascularity in

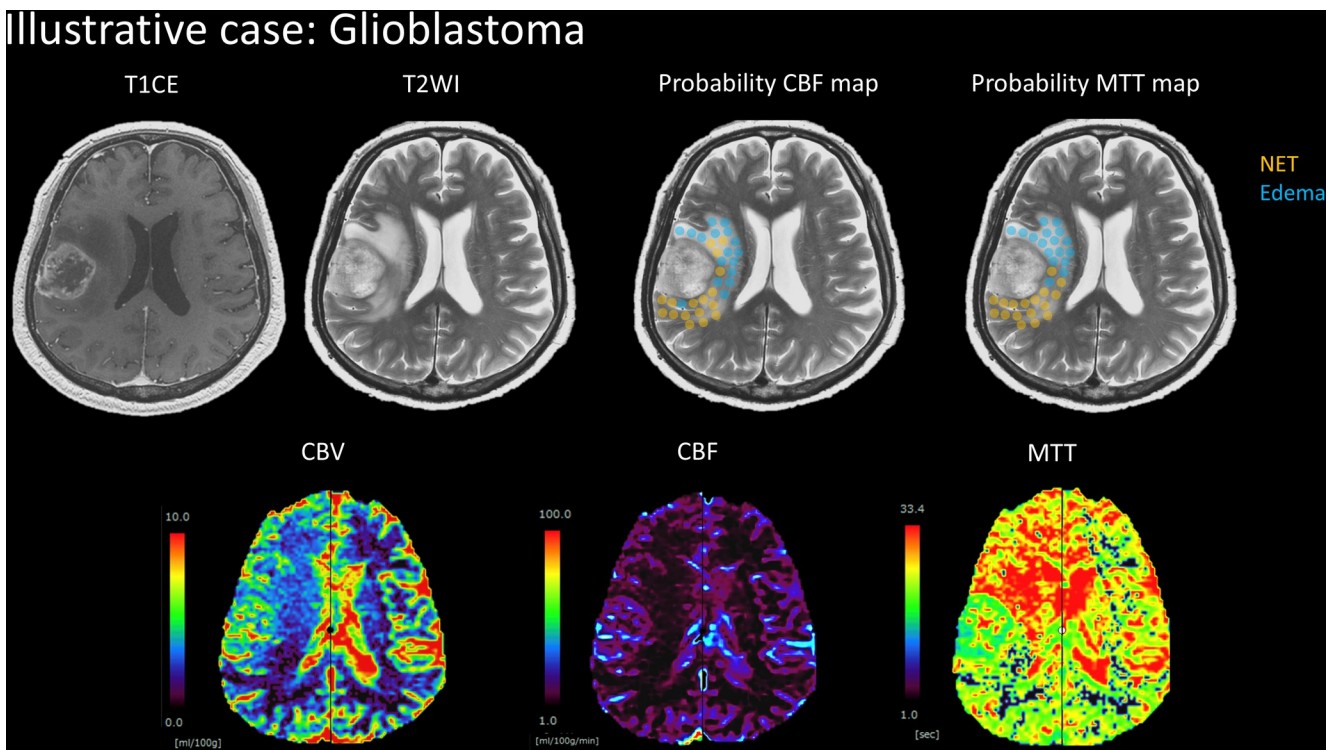

**Fig 9. Illustrative case of glioblastoma.** It shows probability maps for cerebral blood flow (CBF) and mean transit time (MTT), which predict non-contrast-enhancing tumors (NET) and Edema, fused with T2-weighted images. The probability CBF map predicts a NET where the CBF ratio is higher than 0.943, and it predicts Edema where the CBF ratio is lower than 0.943. The probability MTT map predicts a NET where the MTT ratio is lower than 1.229, and it predicts Edema where the MTT ratio is higher than 1.229.

enhancing glioblastoma tumors. They verified that CBV was not significantly correlated with CD31 (p = 0.30) in the core tumor lesions. Furthermore, there was no significant association between Ki-67 and CBV [32]. However, those previous reports investigated the correlation between MR perfusion and only tumor core region, not infiltrative tumor or edema. In our study, the microvessel areas in ETs and NETs were significantly higher than those in Edemas. Moreover, histological evaluation using a 3D scatter plot showed that the spatial distribution of NETs was different from that of Edemas.

CBF and MTT mapping could be used to distinguish between NETs and Edemas on T2-weighted images. Our study showed that the CBF and MTT ratio could predict vascularity in ETs and NETs in patients with glioblastoma. Glioblastoma manifests as diffusely tumor infiltration around pre-existing blood brain vessels and might also induce neovascularization [33], which may have resulted in NETs and ETs showing lower MTT and higher CBF than Edemas. Infiltrative tumor cells induce collective vessel co-option and blood vessel leakage [33], which may result in the progression of NETs to ETs. In our study, the CBF ratios and microvessel areas in ETs and NETs were significantly higher than those in Edemas. Furthermore, the MTT ratio in ETs and NETs were significantly lower than that in Edemas. The vascular structure and permeability of NETs and ETs may involve the process of tumor infiltration and angiogenesis progression.

Based on this histological differentiation between NETs and Edemas, we attempted to radiologically distinguish NETs from Edemas. DSC-MR perfusion images have the potential to discriminate between peritumoral lesions seen in glioblastomas and metastatic brain tumors

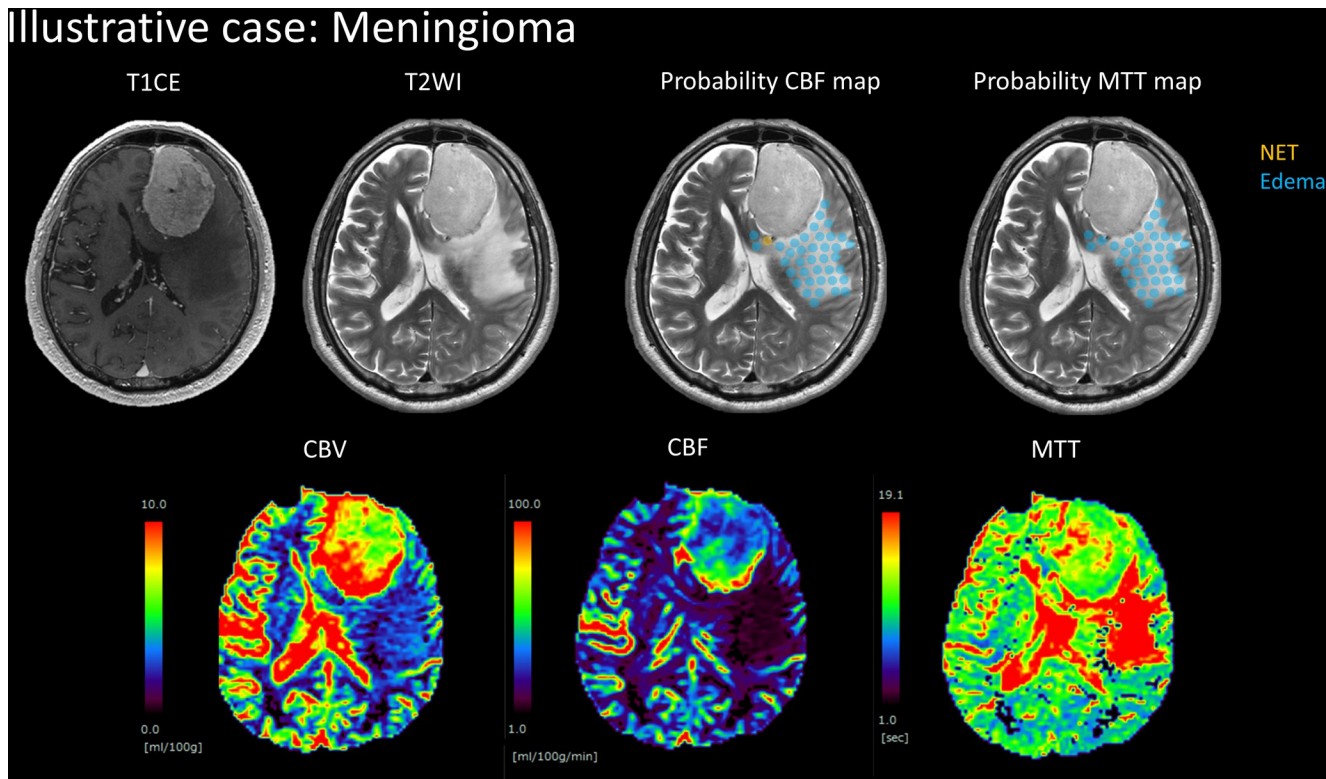

**Fig 10. Illustrative case of meningioma.** It shows probability maps for cerebral blood flow (CBF) and mean transit time (MTT), which predict non-contrast-enhancing tumors (NETs) and Edemas, fused with T2-weighted images. The probability CBF map predicts a NET where the CBF ratio is higher than 0.943, and it predicts Edema where the CBF ratio is lower than 0.943. The probability MTT map predicts NET where the MTT ratio is lower than 1.229, and it predicts Edema where the MTT ratio is higher than 1.229.

cases [13–16]. In these reports, CBV in non-enhancing peritumoral T2-hyperintense region was significantly higher in patients with glioblastoma than that in metastatic brain tumors cases [13–16]. The non-enhancing peritumoral area in glioblastoma in these reports was analyzed without distinguishing between infiltrating tumors and edema. Although a few studies have been conducted to discriminate non-enhancing tumor from perilesional edema using machine learning algorithms with various imaging parameters comprising DSC-MRI, there is still no current consensus on a clinically usable method for reliably differentiating non-enhancing tumor from perilesional edema on perfusion imaging, limiting to small sample size [17–19]. The discrimination between NETs and Edemas using CBV ratio still remains controversial. In our study, CBV ratios in ETs, NETs, and Edemas were not significantly different. We particularly focused on the accurate placement of the ROI for ETs in the solid part of the enhancing tumor area to exclude the confounding effects of tumor heterogeneity, signal loss due to cyst formation, hemorrhage, and large blood vessels. Therefore, the CBV ratio of the ET may not always represent the maximum values. On the other hand, previous reports have determined CBV to be a useful estimation tool for angiogenesis in glioblastoma [34,35]. These reports measured CBV using post-processing with a dedicated software package. However, it is possible that the different software used for image processing of CBV measurements and the varying algorithms may have influenced the results that differ from ours. The methods used for ROI design and imaging processes for CBV may have led to an increased CBF ratio without a significant increase in the CBV ratio in ETs. In addition, we validated the discrimination between NETs and Edemas using ADC values, CBF ratios, and MTT ratios. Notably, previous

reports have shown that ADC values can be used as indicators of glioma proliferation [36,37]. This background supports our investigation, as we demonstrated the potential usefulness of ADC values for discriminating between NETs and Edemas, consistent with the results of CBF and MTT ratios.

Glioblastomas grow aggressively and infiltrate adjacent brain tissue, extending beyond the contrast-enhancing margin. Non-contrast-enhancing lesions beyond the enhancing component of glioblastomas are thought to contain infiltrating tumors and vasogenic edema. According to our radiological analysis, the CBF and MTT ratios for Edemas and Edemas-M showed no significant difference. This suggests that our radiological discrimination between Edemas and NETs on the basis of the previous study [28] might be valid. Distinguishing between NETs and Edemas preoperatively and resecting the infiltrative area contributes to better prognosis and preservation of neurological function. Additionally, visually creating a threshold-based CBF and MTT map can enhance preoperative surgical planning. Moreover, using MRI perfusion preoperatively to differentiate between NETs and Edemas is expected to set an accurate resection margin, allowing for an increased resection rate safely without worsening neurological function. Furthermore, analysis based on local recurrent cases has revealed the potential prognostic value of DSC-MRI, as DSC-MRI parameters can effectively discriminate between recurrent tumors and predict future tumor recurrence [38,39]. Building on this, our study suggests that the CBF ratio and MTT ratio may have the potential to predict recurrent lesions. In particular, the proposed CBF and MTT cut-off values for distinguishing NETs in two patients with glioblastoma and meningioma exhibited potential for differentiating Edemas from infiltrating tumors, albeit with limitations. For instance, the Edema probability CBF map showed the presence of NETs, while the probability MTT map did not identify them, even in the case of meningioma. This suggests that the MTT map was better at detecting edema than the CBF map. Ultimately, a more effective analysis is warranted to determine the accuracy of these cut-off values using $^{11}$C-methionine (MET)-PET, which allows for more accurate delineation of tumor extension than anatomical imaging achieved with MRI [40].

This study faces potential validation challenges. Firstly, it was limited by its small sample size, which could have led to bias in our results. Additionally, we sampled tissue stereotactically using a navigation system, which may have introduced coordinate errors during tissue sampling. Furthermore, we encountered some issues with DSC images. Reproducibility problems with DSC between patients, blurring due to the calculation method, or spatial resolution limitations may also pose difficulties. In the future, the sampling error should be validated while minimizing the effects of brain shifting and standardizing reproducible methods and analysis processes. Overall, future large-scale studies are required to address these methodological challenges and improve the ability to differentiate between NETs and Edemas on T2-weighted images.

There were limitations to this study due to the small sample size and the exploratory nature of the study. Additionally, the low number of specimens may lead to bias in our results, potentially hindering sound statistical analyses. It also remains unclear whether our findings are useful for supratotal resection. Further analysis of the vascular structure and permeability involved in tumor infiltration is necessary to differentiate NETs from Edemas in terms of tumor infiltration mechanisms. Furthermore, large-scale studies would enhance statistical power and validate the proposed differentiation between NETs and Edemas on T2-weighted images from a prognostic perspective.

In conclusion, we aimed to differentiate between NETs and Edemas in glioblastoma using DSC-MR perfusion imaging. Performing MR perfusion imaging in addition to conventional MRI may prove useful in differentiating infiltrating tumors from vasogenic edema in non-contrast T2 hyperintensity regions of glioblastoma.

## Author Contributions

**Conceptualization:** Yoshiko Okita, Manabu Kinoshita.

**Data curation:** Yoshiko Okita, Reina Utsugi, Koki Murakami, Ryuichi Hirayama, Noriyuki Kijima, Hideyuki Arita, Manabu Kinoshita, Yasunori Fujimoto, Hajime Nakamura, Naoki Kagawa.

**Formal analysis:** Hideki Kuroda.

**Funding acquisition:** Haruhiko Kishima.

**Investigation:** Yoshiko Okita.

**Methodology:** Yoshiko Okita, Atsuko Arisawa, Noriyuki Tomiyama.

**Project administration:** Yoshiko Okita.

**Resources:** Haruhiko Kishima.

**Supervision:** Noriyuki Tomiyama, Haruhiko Kishima.

**Validation:** Hideki Kuroda, Yoshiko Okita.

**Visualization:** Yoshiko Okita, Atsuko Arisawa.

**Writing – original draft:** Hideki Kuroda, Yoshiko Okita.

**Writing – review & editing:** Yoshiko Okita, Atsuko Arisawa, Haruhiko Kishima.

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
