## [Decision Letter · Decision Letter 0]

17 Sep 2024

PONE-D-24-36230Cerebral blood flow and histological analysis for the accurate differentiation of infiltrating tumor and vasogenic edema in glioblastomaPLOS ONE

Dear Dr. Okita,

Thank you for submitting your manuscript to PLOS ONE. After careful consideration, we feel that it has merit but does not fully meet PLOS ONE’s publication criteria as it currently stands. Therefore, we invite you to submit a revised version of the manuscript that addresses the points raised by reviewers 1, 2 and 3 during the review process.

We look forward to receiving your revised manuscript.

Kind regards,

Pradeep Kumar, Ph.D.

Academic Editor

PLOS ONE

**Journal Requirements:**

https://www.sciencedirect.com/science/article/abs/pii/S0730725X23000474?via%3Dihub

In your revision ensure you cite all your sources (including your own works), and quote or rephrase any duplicated text outside the methods section. Further consideration is dependent on these concerns being addressed.

This work was supported by JST Grant Number JPMJPF2009.The funders had no role in study design, data collection and analysis, decision to publish, or preparation of the manuscript.

4. In the online submission form, you indicated that The data generated in the present study are not publicly available due to them containing information that could compromise research participant privacy/consent but may be requested from the corresponding author.

5. Please upload a new copy of Figures 2, 3 and 4 as the detail is not clear. Please follow the link for more information: " ext-link-type="uri" xlink:type="simple">https://blogs.plos.org/plos/2019/06/looking-good-tips-for-creating-your-plos-figures-graphics/"
" ext-link-type="uri" xlink:type="simple">https://blogs.plos.org/plos/2019/06/looking-good-tips-for-creating-your-plos-figures-graphics/"

6. We note you have included a table to which you do not refer in the text of your manuscript. Please ensure that you refer to Table 1 in your text; if accepted, production will need this reference to link the reader to the Table.

Reviewers' comments:

Reviewer's Responses to Questions

**Comments to the Author**

1. Is the manuscript technically sound, and do the data support the conclusions?

Reviewer #1: Yes

Reviewer #2: Yes

Reviewer #3: Yes

2. Has the statistical analysis been performed appropriately and rigorously? 

Reviewer #1: Yes

Reviewer #2: No

Reviewer #3: Yes

3. Have the authors made all data underlying the findings in their manuscript fully available?

Reviewer #1: Yes

Reviewer #2: Yes

Reviewer #3: Yes

4. Is the manuscript presented in an intelligible fashion and written in standard English?

Reviewer #1: Yes

Reviewer #2: Yes

Reviewer #3: Yes

5. Review Comments to the Author

**Reviewer #1:** Peer Review Report: Analysis of Cerebral Blood Flow and Histopathology for Differentiation of Infiltrating Tumor and Vasogenic Edema in Glioblastoma

1. Summary of the Study

This manuscript examines the application of dynamic susceptibility contrast (DSC) MRI to distinguish non-contrast-enhancing tumors (NETs) from vasogenic edema in glioblastoma. Using cerebral blood flow (CBF), cerebral blood volume (CBV), and mean transit time (MTT) measurements from MRI, coupled with histological data on cell density, Ki-67 index, and microvessel area, the study aims to enhance diagnostic precision. The results highlight significant differences in cell density and vascular characteristics between NETs and edemas, with CBF and MTT being particularly effective in differentiating these regions.

2. Strengths of the Study

Clinical Significance: The study addresses a key issue in the treatment of glioblastoma—accurately differentiating between tumor infiltration and edema. This distinction is vital for improving surgical strategies and overall patient care.

Comprehensive Approach: The study effectively combines advanced imaging techniques with histological validation, lending greater credibility to the findings. The use of stereotactic biopsies ensures accurate correlation between imaging data and tissue pathology.

Innovative Focus: By concentrating on the non-contrast-enhancing regions of glioblastomas, the research tackles a challenging and clinically relevant area that has implications for treatment planning and outcome prediction.

3. Major Concerns

Small Sample Size: The relatively small number of patients (48 with glioblastoma and 24 with meningioma as controls) may limit the robustness and generalizability of the results. Larger studies are necessary to confirm the findings and broaden their applicability.

Limited Discussion on Clinical Impact: Although the study demonstrates the utility of CBF and MTT for differentiating NETs and edemas, there is limited discussion on how these findings could translate into clinical decision-making, particularly in terms of surgical planning or modifying treatment strategies.

Lack of Comparison with Other Modalities: The study would benefit from comparing DSC-MRI with other imaging techniques, such as diffusion-weighted imaging or arterial spin labeling, to better contextualize the advantages and limitations of DSC-MRI in distinguishing tumor from edema.

4. Minor Issues

Clarity of Data Presentation: Some tables and figures, particularly those depicting imaging data, could benefit from more detailed legends and explanations to improve interpretability (e.g., Figure 4).

ROC Analysis Application: While the ROC analysis for CBF and MTT is statistically strong, further elaboration on how these metrics might be applied in clinical settings would enhance the study's practical relevance.

Control Group Explanation: The inclusion of meningioma patients as controls for vasogenic edema is valuable; however, a more thorough explanation of their selection and relevance to glioblastoma would strengthen the rationale for using this group as a comparison.

5. Suggestions for Improvement

Larger Sample Sizes in Future Studies: Expanding the cohort size in future studies would improve the statistical power and help validate the findings across a broader population of glioblastoma patients.

Incorporation of Prognostic Data: Adding data on clinical outcomes, such as patient survival and progression-free intervals, would provide a more comprehensive understanding of the clinical significance of distinguishing NETs from edema using DSC-MRI.

Comparison with Other Imaging Techniques: Including a comparison with alternative imaging modalities could highlight the specific advantages of DSC-MRI and provide a more balanced view of its role in differentiating NETs from edema.

6. Conclusions and Recommendations

The manuscript offers valuable insights into the use of DSC-MRI for distinguishing NETs from vasogenic edemas in glioblastoma. The integration of imaging data with histopathological analysis is a significant strength of the study, and the findings have the potential to impact clinical management. However, the small sample size and limited discussion on clinical applications slightly reduce the study’s overall impact. Expanding the sample size and incorporating additional outcome data would further enhance the value of the research.

Recommendation: The manuscript is a strong candidate for publication, provided revisions are made to address the issues related to sample size, data presentation, and the practical implications of the findings for clinical practice. With these improvements, the study could make a substantial contribution to the field of glioblastoma imaging and treatment.

**Reviewer #2:** This is an interesting study and the authors have collected a unique dataset using cutting edge methodology.

This paper has good conception and validation in crucial decisions for glioblastoma appraisal. The paper is generally clear and structured. Sufficient information about this study findings is presented for readers to follow the present study rationale and procedures in the flowchart/data processing (Figure 1). Notwithstanding, in my opinion, the paper has some imperfections concerning some data analyses and text, and this unique dataset has not been availed to its full extent.

Key critical points are:

Quantitative and qualitative research for data type approach in statistical results and analysis is important. Especially, Figure 1 is the conception of this study in data analysis. Unfortunately, The reviewer didn't see any tables describing it in sufficient detail.

(1) The biggest issue is that the data analysis should more robust and so, the authors are going to have to find a way to properly demonstrate their data. How about the power analysis in statistics?

(2) The image mark-up plays an important role in results. The Figure 1 is divided into three colors marked segments. What is your groundwork based on these color-marked ?

Ki-67 labeling index is an important indicator of tumor cell proliferation in glioma, which can only be obtained by postoperative biopsy regarding microvascular proliferation as a crucial histological feature of glioblastoma.

(3) Could you show the different ROIs correspondingly in histological features in this study? Additionally, please discuss this point in the Discussion section.

Data Implement

(4) Regarding the edema, T2*-based dynamic susceptibility contrast (DSC) MRI is a good point for approach, however, correlation coefficient value interpretation is weak for low correlation. Could you discuss this part in the discussion?

(5) What else about the study limitations except a small sample size? Please add more in Discussion.

**Reviewer #3:** PLOS ONE

Ms type: Research Article

Ms ID: PONE-D-24-36230

Title: Cerebral blood flow and histological analysis for the accurate differentiation of infiltrating tumor and vasogenic edema in glioblastoma

General Comments:

The study presents valuable findings regarding the use of DSC-MRI perfusion imaging for differentiating non-contrast-enhancing tumors (NETs) from vasogenic edema in glioblastomas. The correlation observed between CBF and histopathological parameters such as cell density, Ki-67 index, and microvessel area is particularly noteworthy. However, a major point of concern arises from the lack of correlation between CBV and these same parameters, despite the fact that several studies in the literature suggest that CBV is typically more sensitive than CBF in reflecting neovascularization, especially in gliomas.

1. Explanation for the Discrepancy between CBF and CBV: It would be beneficial for the authors to provide a more in-depth explanation or hypothesis as to why CBV did not show a significant correlation with microvessel area, cell density, or Ki-67 index, while CBF did. Given that both CBF and CBV are considered reliable markers of neovascularization, the observed discrepancy raises important questions. In some studies, CBV has been shown to better capture neovascularization than CBF. The authors should address whether the lack of correlation could be due to technical factors, such as the selection of arterial input function (AIF) or the specific post-processing algorithms used in the study, particularly since CBV calculation is known to be more sensitive to such variables.

2. The Role of Vessel Co-option: Additionally, the authors may wish to consider the role of vessel co-option, which is prominent in glioblastomas. Vessel co-option could explain why CBF increases without a corresponding rise in CBV, as pre-existing vasculature is co-opted by infiltrating tumor cells. This phenomenon may result in increased blood flow (CBF) without a significant increase in vessel volume (CBV), particularly in the NET regions. A discussion on whether this mechanism could account for the differential behavior of CBF and CBV would provide valuable context to the findings.

3. Potential Influence of Tumor Stage or Region: It may also be worthwhile to discuss whether the stage of tumor progression or the specific tumor region examined could influence the sensitivity of CBF versus CBV in detecting neovascularization. Early-stage tumors or regions with active infiltration might show different perfusion characteristics than more established tumor regions. Clarifying whether the study accounted for such factors would strengthen the interpretation of the results.

In summary, while the results of the study are promising, addressing the discrepancy between CBF and CBV and providing a more comprehensive discussion on possible reasons for this observation would greatly enhance the robustness and clinical relevance of the findings.

Minor Comments:

4. Presentation of Representative Images: To further enhance the clarity and impact of the study, it would be valuable to include representative images from both glioblastoma and meningioma cases. Specifically, providing clear examples of DSC-MR perfusion images with corresponding CBF, CBV, and MTT maps for typical cases of NETs and Edemas in glioblastoma, as well as for vasogenic edema in meningioma, would help readers better understand the radiological differences described in the study. These images would also visually support the conclusions drawn from the quantitative data and improve the overall comprehensibility of the findings.

5. Inclusion of Color Scale Units and LUT Bars in Figure 1: A minor but important detail for improving the clarity of Figure 1 is the addition of the color scale units and LUT (Lookup Table) bars for the CBV, CBF, and MTT maps. Including these will allow readers to interpret the quantitative perfusion values more effectively. Clear color scales and units are essential for accurately assessing the radiological findings presented and ensuring that the visual data aligns with the statistical analysis discussed in the manuscript.

6. PLOS authors have the option to publish the peer review history of their article (what does this mean?). If published, this will include your full peer review and any attached files.

Reviewer #1: No

Reviewer #2: No

Reviewer #3: No

---

## [Author Response · Author response to Decision Letter 0]

15 Nov 2024

Reviewer #1: 

3. Major Concerns

3.1. Small Sample Size: The relatively small number of patients (48 with glioblastoma and 24 with meningioma as controls) may limit the robustness and generalizability of the results. Larger studies are necessary to confirm the findings and broaden their applicability.

Answer: We thank you for your comment. We agree with your concern and have addressed it in the limitation section. This study was limited by its small sample size, which could lead to potential bias in our results. Therefore, future large-scale studies are required to validate the differentiation of infiltrating tumors and vasogenic edema with MRI perfusion. We have added the following statement to the revised manuscript (page 17, line 18): This study faces potential validation challenges. Firstly, it was limited by its small sample size, which could have led to bias in our results. Additionally, we sampled tissue stereotactically using a navigation system, which may have introduced coordinate errors during tissue sampling. Furthermore, we encountered some issues with DSC images. Reproducibility problems with DSC between patients, blurring due to the calculation method, or spatial resolution limitations may also pose difficulties. In the future, the sampling error should be validated while minimizing the effects of brain shifting and standardizing reproducible methods and analysis processes. Overall, future large-scale studies are required to address these methodological challenges and improve the ability to differentiate between NETs and Edemas on T2-weighted images.

3.2. Limited Discussion on Clinical Impact: Although the study demonstrates the utility of CBF and MTT for differentiating NETs and edemas, there is limited discussion on how these findings could translate into clinical decision-making, particularly in terms of surgical planning or modifying treatment strategies.

Answer: We thank you for your valuable comment. As you noted, this is an important point. Improving the extent of resection in the surgical treatment of glioblastoma leads to a better prognosis. Although the utility of supramarginal resection needs further validation, it is expected to contribute to improved outcomes. The T2 high-intensity lesion surrounding the contrast-enhanced lesion contains a mixture of edema and infiltrative tumor. Distinguishing between the NETs and Edemas preoperatively and resecting the infiltrative area contributes to better prognosis and preservation of neurological function. Using MRI perfusion preoperatively to differentiate between NETs and Edemas will be expected to set an accurate resection margin that will allow for an increase in the resection rate safely without worsening neurological function. Accordingly, we have added the following statement to the revised manuscript (page 16, line 21): Distinguishing between NETs and Edemas preoperatively and resecting the infiltrative area contributes to better prognosis and preservation of neurological function. Additionally, visually creating a threshold-based CBF and MTT map can enhance preoperative surgical planning. Moreover, using MRI perfusion preoperatively to differentiate between NETs and Edemas is expected to set an accurate resection margin, allowing for an increased resection rate safely without worsening neurological function.

3.3. Lack of Comparison with Other Modalities: The study would benefit from comparing DSC-MRI with other imaging techniques, such as diffusion-weighted imaging or arterial spin labeling, to better contextualize the advantages and limitations of DSC-MRI in distinguishing tumor from edema.

Answer: We thank you for your insightful comment. As you highlighted, this is an important point. We validated the differentiation of NET from Edema using DSC-MRI and ADC values. The ADC ratios were significantly higher in NETs than in Edemas (p0.05). However, no significant difference was observed between ETs and NETs in ADC ratios. Accordingly, we have added the following statement to the revised manuscript (page 11, line 9): 

No significant difference was observed in the mean CBV ratios between Edemas, NETs, and ETs (Figure 3 D). The mean CBF ratio for Edemas was significantly lower than that for NETs (p0.01). In contrast, the mean MTT ratio for Edemas was significantly higher than that for NETs (p0.01). However, no significant difference was observed in the CBF and MTT ratios between ETs and NETs (Figure 3 E and F). To further clarify the distinction between NETs and Edemas, we compared the validation using ADC values. The ADC ratios were significantly higher in NETs than in Edemas (p0.05) (Figure 3 G). However, no significant difference was observed in ADC ratios between ETs and NETs.

We have added the following statement to the revised manuscript (page 16, line 8): 

In addition, we validated the discrimination between NETs and Edemas using ADC values, CBF ratios, and MTT ratios. Notably, previous reports have shown that ADC values can be used as indicators of glioma proliferation [36, 37]. This background supports our investigation, as we demonstrated the potential usefulness of ADC values for discriminating between NETs and Edemas, consistent with the results of CBF and MTT ratios.

We also added the following statement to the revised manuscript (page 7, line 1): 

DTI was acquired using a single-shot echo planar imaging technique with TE=80 and TR=10,000. Diffusion gradient encoding in 25 directions with b=2,000 s/mm2 and an additional measurement without the diffusion gradient (b=0 s/mm2) was performed [23].

We also added the following statement to the revised manuscript (page 7, line 12): 

DTI was also acquired qualitatively apparent diffusion coefficient (ADC) values using an ADC map.

We also added the following statement to the revised manuscript (page 21, line 17): Figure 3

D, E, F, and G

 Boxplots of the cerebral blood volume (CBV), cerebral blood flow (CBF), mean transit time (MTT), and apparent diffusion coefficient (ADC) ratios in non-contrast-enhancing tumors (NETs), edemas (Edemas), and enhancing tumors (ETs) based on a histological comparison using stereotactic imaging. The mean CBF ratio for Edemas was significantly lower than that for NETs (p0.01). The mean MTT ratio for Edemas was significantly higher than that for NETs (p0.01) The ADC ratios were significantly higher in NETs than in Edemas (p0.05). However, no significant difference was observed in CBV ratios between NETs and Edemas.

(page 22, line 13): 

Figure 5

Correlation of the ADC ratio with cell density, Ki-67, and microvessel area based on a histological comparison using stereotactic imaging. The ADC ratio shows no correlation with cell density, the Ki-67 index, or microvessel area.

4. Minor Issue

4.1. Clarity of Data Presentation: Some tables and figures, particularly those depicting imaging data, could benefit from more detailed legends and explanations to improve interpretability (e.g., Figure 4).

Answer: We thank you for your comment. We agree with your comment. Accordingly, we have added detailed legends and explanations. We have also added the following statement to the revised manuscript (page 21, line 7): 

Figure3

A, B, and C

 Boxplots of the Ki-67 index, cell density, and microvessel area in non-contrast-enhancing tumors (NETs), edemas (Edemas), and enhancing tumors (ETs). The Ki-67 index, cell density, and microvessel area were significantly higher in NETs compared to Edemas (p0.05, p0.01, and p0.05, respectively). The Ki-67 index and microvessel area were significantly higher in ETs than in NETs (p0.05 and p0.05, respectively). However, no significant difference in cell density between ETs and NETs was observed. 

D, E, F, and G

 Boxplots of the cerebral blood volume (CBV), cerebral blood flow (CBF), mean transit time (MTT), and apparent diffusion coefficient (ADC) ratios in non-contrast-enhancing tumors (NETs), edemas (Edemas), and enhancing tumors (ETs) based on a histological comparison using stereotactic imaging. The mean CBF ratio for Edemas was significantly lower than that for NETs (p0.01). The mean MTT ratio for Edemas was significantly higher than that for NETs (p0.01). The ADC ratios were significantly higher in NETs than in Edemas (p0.05). However, no significant difference was observed in CBV ratios between NETs and Edemas.

(page 23, line 3): 

Figure 7

Boxplots of the cerebral blood volume (CBV), cerebral blood flow (CBF), and mean transit time (MTT) ratios in non-contrast-enhancing tumors (NETs), edemas (Edemas), enhancing tumors (ETs) in 48 glioblastoma patients, and Edemas in the control group (Edemas-M). The mean CBF ratio for Edemas (0.56; range, 0.11–1.08) was significantly lower than that for NETs (1.64; range, 0.39–4.05) (p0.01). The mean MTT ratio for Edemas (1.83; range, 1.22–3.30) was significantly higher than that for NETs (0.96; range, 0.34–1.60) (p0.01). In contrast, the CBF and MTT ratios for Edemas and the controls exhibited similar tendencies.

Boxplots of the cerebral blood flow (CBF) and mean transit time (MTT) ratios in local recurrent tumors in 12 relapsed glioblastoma patients. The mean CBF ratio for the recurrent tumor was 1.136, exceeding the cut-off value of 0.943 to predict NETs. The mean MTT ratio for the recurrent tumor was 0.944, falling below the cut-off value of 1.229 to predict NETs.

(page 23, line 19):

Figure 8

The receiver operating characteristic (ROC) curve shows reliable predictions that distinguish non-contrast-enhancing tumors (NETs) and edemas (Edemas) in glioblastoma in terms of cerebral blood flow (CBF) and mean transit time (MTT). The CBF ratio (area under the curve [AUC] =0.890) effectively distinguishes between NETs and Edemas, with a sensitivity of 74.2% and a specificity of 97.1% (cut-off value =0.943, p0.01). The MTT ratio (AUC=0.946) also effectively distinguishes between NETs and Edemas, with a sensitivity of 80.6% and a specificity of 97.1% (cut-off value =1.229, p0.01).

4.2. ROC Analysis Application: While the ROC analysis for CBF and MTT is statistically strong, further elaboration on how these metrics might be applied in clinical settings would enhance the study's practical relevance.

Answer: We thank you for your comment. As you noted, this is an important point. Accordingly, we have added the following statement to the revised manuscript (page 13, line 5): We presented the probability map applied to NETs and Edemas lesions in two patients with glioblastoma and meningioma (Figures 9 and 10). The probability map predicting NETs indicated a CBF ratio over 0.943, represented in orange. The area colored blue corresponded to Edema CBF ratios below 0.943. Additionally, the probability map predicting NETs showed an MTT ratio below 1.229, also colored orange, while the area colored blue applied to Edema MTT ratios above 1.229. Notably, the area of the probability CBF map for Edemas containing NETs did not necessarily align with the area in the probability MTT map for Edemas in both glioblastoma and meningioma cases.

We also have added the following statement to the revised manuscript (page 17, line 8): In particular, the proposed CBF and MTT cut-off values for distinguishing NETs in two patients with glioblastoma and meningioma exhibited potential for differentiating Edemas from infiltrating tumors, albeit with limitations. For instance, the Edema probability CBF map showed the presence of NETs, while the probability MTT map did not identify them, even in the case of meningioma. This suggests that the MTT map was better at detecting edema than the CBF map. Ultimately, a more effective analysis is warranted to determine the accuracy of these cut-off values using 11C-methionine (MET)-PET, which allows for more accurate delineation of tumor extension than anatomical imaging achieved with MRI[40].

We also have added the following statement to the revised manuscript (page 24, line 5):

 Figure 9

Illustrative case of glioblastoma. It shows probability maps for cerebral blood flow (CBF) and mean transit time (MTT), which predict non-contrast-enhancing tumors (NET) and Edema, fused with T2-weighted images. The probability CBF map predicts a NET where the CBF ratio is higher than 0.943, and it predicts Edema where the CBF ratio is lower than 0.943. The probability MTT map predicts a NET where the MTT ratio is lower than 1.229, and it predicts Edema where the MTT ratio is higher than 1.229.

Figure 10

Illustrative case of meningioma. It shows probability maps for cerebral blood flow (CBF) and mean transit time (MTT), which predict non-contrast-enhancing tumors (NETs) and Edemas, fused with T2-weighted images. The probability CBF map predicts a NET where the CBF ratio is higher than 0.943, and it predicts Edema where the CBF ratio is lower than 0.943. The probability MTT map predicts NET where the MTT ratio is lower than 1.229, and it predicts Edema where the MTT ratio is higher than 1.229.

4.3. Control Group Explanation: The inclusion of meningioma patients as controls for vasogenic edema is valuable; however, a more thorough explanation of their selection and relevance to glioblastoma would strengthen the rationale for using this group as a comparison.

Answer: We thank you for your insightful feedback. Accordingly, we have added the flowchart for the case accumulation of glioblastoma and meningioma as controls. We have also added Figure 1 to outline the criteria used in this study. We have added the following statement to the revised manuscript (page 20, line 5): 

Figure 1

The process of patient selection for inclusion in the study. We retrospectively collected the medical records of 75 patients diagnosed with IDH-wildtype glioblastoma between January 2017 and January 2023 at our institution. Of those 75 patients, 27 were excluded from the study due to tumors with unsatisfactory images and/or prior radiotherapy before MRI examination, leaving 48 eligible for radiological analysis. In 14 of these cases, stereotactic sampling was performed. During the review period, local recurrent analysis was also conducted. Of the 48 patients, 36 were excluded from the recurrent analysis due to recurrent tumors with unsatisfactory images, biopsy or partial resection, lack of recurrence, or no follow-up. Additionally, we retrospectively collected the medical records of 58 patients diagnosed with meningioma between January 2017 and January 2023 at our institution. Of those 58 patients, 38 were excluded from the study due to tumors with unsatisfactory images, Grade 2/3 meningioma, or infratentorial tumors.

5. Suggestions for Improvement

5.1. Larger Sample Sizes in Future Studies: Expanding the cohort size in future studies would improve the statistical power and help validate the findings across a broader population of glioblastoma patients.

Answer: We thank you for your comment. We agree with your suggestion. This study was limited by its small sample size. This is an exploratory study, and whether this study’s results are universal remains unclear. Future large-scale studies are required to validate the differentiation of infiltrating tumors and vasogenic edema with MRI perfusion. Accordingly, we have added the following statement to the revised manuscript (page 18, line 5): There were limitations to this study due to the small sample size and the exploratory nature of the study. Additionally, the low number of specimens may lead to bias in our results, potentially hindering sound statistical analyses. It also remains unclear whether our findings are useful for supratotal resection. Further analysis of the vascular structure and permeability involved in tumor infiltration is necessary to differentiate NETs from Edemas in terms of tumor infiltration mechanisms. Furthermore, large-scale studies would enhance statistical power and validate the proposed differentiation between NETs and Edemas on T2-weighted images from a prognostic perspective.

5.2. Incorporation of Prognostic Data: Adding data on clinical outcomes, such as patient survival and progression-free intervals, would provide a more comprehensive understanding of the clinical significance of distinguishing NETs from edema using DSC-MRI.

Answer: We thank you for your valuable comment. As you noted, this is an im

---

## [Decision Letter · Decision Letter 1]

9 Dec 2024

Cerebral blood flow and histological analysis for the accurate differentiation of infiltrating tumor and vasogenic edema in glioblastoma

PONE-D-24-36230R1

Dear Dr. Okita,

We’re pleased to inform you that your manuscript has been judged scientifically suitable for publication and will be formally accepted for publication once it meets all outstanding technical requirements.

Kind regards,

Pradeep Kumar, Ph.D.

Academic Editor

PLOS ONE

Additional Editor Comments (optional):

Reviewers' comments:

Reviewer's Responses to Questions

**Comments to the Author**

1. If the authors have adequately addressed your comments raised in a previous round of review and you feel that this manuscript is now acceptable for publication, you may indicate that here to bypass the “Comments to the Author” section, enter your conflict of interest statement in the “Confidential to Editor” section, and submit your "Accept" recommendation.

Reviewer #2: All comments have been addressed

Reviewer #3: All comments have been addressed

2. Is the manuscript technically sound, and do the data support the conclusions?

Reviewer #2: Yes

Reviewer #3: Yes

3. Has the statistical analysis been performed appropriately and rigorously? 

Reviewer #2: Yes

Reviewer #3: Yes

4. Have the authors made all data underlying the findings in their manuscript fully available?

Reviewer #2: Yes

Reviewer #3: Yes

5. Is the manuscript presented in an intelligible fashion and written in standard English?

Reviewer #2: Yes

Reviewer #3: Yes

6. Review Comments to the Author

Reviewer #2: (1) Interesting, What software should make Fig. 6 Representation of the correlation between cell density, Ki-67 index, and microvessel area in ET, NET, and edema for stereotaxic assessment?

(2) Much Better! All comments have been addressed. Thank you for all your hard work!

Reviewer #3: The authors responded to the comments satisfactorily, and the changes improved the quality of the text.

7. PLOS authors have the option to publish the peer review history of their article (what does this mean?). If published, this will include your full peer review and any attached files.

Reviewer #2: No

Reviewer #3: No

---

## [Editor Report · Acceptance letter]

29 Dec 2024

PONE-D-24-36230R1 

PLOS ONE

Dear Dr. Okita, 

I'm pleased to inform you that your manuscript has been deemed suitable for publication in PLOS ONE. Congratulations! Your manuscript is now being handed over to our production team.

Kind regards, 

on behalf of

Prof. Pradeep Kumar 

Academic Editor

PLOS ONE